# OpenReview forum: "CoT is Not the Chain of Truth: An Empirical Internal Analysis of Reasoning LLMs for Fake News Generation"
_ICML.cc/2026/Conference — ICML 2026 regular_

### Official Review · Reviewer_U8k4 · 2026-03-01

**Soundness:** 3
**Presentation:** 3
**Significance:** 3
**Originality:** 3
**Overall Recommendation:** 5
**Confidence:** 4

**Summary:**

This paper challenges the widely held assumption that a Large Language Model (LLM)’s refusal of harmful fake news generation requests indicates fully safe internal Chain-of-Thought (CoT) reasoning, revealing that ~80% of such refusal cases still contain latent unsafe reasoning traces in CoT. The authors design a unified safety-analysis framework to deconstruct CoT generation from layers to attention heads, first localizing safety-critical mid-depth contiguous layers via representation separation analysis, then introducing a Jacobian-based spectral method with three interpretable metrics (stability, geometry, energy) to identify safety-critical attention heads within these layers. They construct a labeled CoT dataset for fake news generation (FNG) tasks with direct/indirect prompting and stylistic conditioning, conduct extensive experiments on Llama-8B, Qwen-4B, and Qwen-8B, and validate the causal link between critical routing components and CoT safety via targeted perturbation. The work provides a mechanistic understanding of unsafe CoT reasoning and a fine-grained localization method for mitigating latent reasoning risks in LLMs. Overall, a central concept investigated by this article is the disconnect between surface-level LLM output compliance (refusal of harmful requests) and the latent unsafe reasoning patterns embedded in the internal CoT process for FNG tasks, and how to systematically identify and measure these unsafe patterns through mechanistic interpretability of attention routing. This article's significant contribution pertains to the development of a coarse-to-fine Jacobian-based spectral analysis framework that enables precise localization of safety-critical layers and attention heads, along with three physics-inspired metrics that transform abstract CoT safety into measurable routing properties for FNG-specific LLM reasoning risk mitigation.

**Compliance With Llm Reviewing Policy:**

Affirmed.

**Final Justification:**

The authors have addressed my concerns, so I raise my score to 5.

**Key Questions For Authors:**

1. Have you tested the proposed framework and spectral metrics on larger LLMs (e.g., 70B scale) or models with different alignment strategies (RLHF/DPO)? If not, what challenges do you anticipate in generalizing the findings to these models, and what preliminary steps could address these challenges?
2. Can you outline 1-2 concrete, implementable intervention strategies (e.g., fine-tuning regularizers, real-time monitoring pipelines) that leverage the identified safety-critical layers/heads and spectral metrics, and provide preliminary results on their effectiveness in reducing unsafe CoT reasoning?
3. How do the computational costs of the Jacobian-based spectral analysis framework compare to state-of-the-art CoT monitoring methods (e.g., attention heatmapping, self-evaluation faithfulness metrics)? Are there any model compression or approximation techniques that could reduce the computational burden for real-time LLM safety monitoring?
4. The paper notes that indirect prompting shifts safety-critical layers deeper—what mechanistic explanations can you provide for this phenomenon, and do these explanations translate to other types of adversarial prompting (e.g., few-shot jailbreak prompts)?

**Limitations:**

yes

**Strengths And Weaknesses:**

## Strengths
1. The paper uncovers a critical gap in existing LLM safety research—output-level refusal does not equate to internal reasoning safety—with empirical evidence showing ~80% of refusal cases have unsafe CoT traces across three LLMs. This insight directly challenges foundational safety assumptions and addresses a high-stakes problem in LLM misinformation mitigation, with experiments on diverse model architectures/scales (Llama-8B, Qwen-4B/8B) and prompting/stylistic settings ensuring generalizability of findings.
2. The coarse-to-fine analysis pipeline (layer localization → attention head characterization) is methodologically rigorous. The layer localization uses cosine similarity-based representation separation to identify contiguous safety-critical mid-depth layers, and the Jacobian-based spectral analysis of the softmax operator is a novel and principled approach to quantify attention routing behavior—avoiding the limitations of attention heatmaps that only visualize outcomes. The three derived metrics (stability, geometry, energy) are well-defined with theoretical proofs (Appendix E-F) and provide complementary, interpretable measures of routing safety.
3. The authors construct a specialized labeled CoT dataset for FNG with clear safety taxonomies (Safe, Potential Unsafe, Unsafe) and a systematic annotation process (three annotators, cross-validation, LLM-assisted rule formulation) that ensures data quality. The targeted anti-direction perturbation experiments on critical layers/heads provide causal evidence for the link between routing properties and CoT safety, demonstrating that perturbing critical components leads to far more severe safety rate degradation than random perturbations—strengthening the paper’s core claims about the functional role of these components.

## Weaknesses
1. The experiments are only conducted on three relatively small LLMs (8B/4B scale) with decoder-only transformer architectures. There is no analysis of larger models (e.g., 70B scale), encoder-decoder models, or open-source/closed models with different alignment strategies (e.g., RLHF, DPO). It is unclear whether the identified safety-critical layer/head localization patterns and spectral metrics generalize to these broader model classes, which limits the real-world applicability of the findings.
2. While the paper excels at identifying safety-critical routing components, it provides only high-level suggestions for "targeted interventions" without concrete, implementable strategies (e.g., pruning critical heads, regularizing their spectral properties during fine-tuning, real-time monitoring of metric values). The lack of practical intervention experiments means the work remains largely descriptive, with an unaddressed gap between identifying risky components and mitigating the risks in deployed LLMs.
3. The related work section discusses CoT monitoring approaches (self-evaluation, external-supervision) but the paper does not include a quantitative comparison between its Jacobian-based framework and these state-of-the-art methods (e.g., in terms of CoT safety detection accuracy, computational efficiency). This makes it difficult to assess the incremental improvement of the proposed framework over existing techniques, a key measure of its practical value.

---

> ### Author Rebuttal · Authors · 2026-03-31
>
> ①U8k4-W1:Limited model scale and architecture diversity in experiments.
>
> Thank you for your insightful suggestions. To address your concern regarding the broad applicability of our findings,We extended our evaluation to two additional representative models: Flan-UL2 (20B, encoder-decoder) and DeepSeek-R1-70B (70B, aligned model). The same core pattern still holds: safety-relevant signals concentrate in a few critical layers and further aggregate onto a small subset of critical heads. Specifically, concentration(Conc.) measures how much of the total gap is captured by critical components, and Relative Gap measures how much stronger the average gap on critical heads is than on ordinary heads. These results support the generalizability of our framework across larger scales, different architectures, and different alignment paradigms.
>
> | Model | Total  | News Layer Range | News Conc. |
> |---|---:|---|---:|
> | Llama-8B | 32 | [18, 20] | 73.1% |
> | Qwen-4B | 36 | [21, 23] | 71.9% |
> | Qwen-8B | 36 | [22, 24] | 72.3% |
> | Flan-UL2-20B | 32 | [28, 30] | 88.9% |
> | DeepSeek-R1-70B | 80 | [65, 67] | 64.9% |
>
> | Model | B1 Relative Gap | B2 Relative Gap | B3 Relative Gap |
> |---|---:|---:|---:|
> | Llama-8B | 321.4% | 322.6% | 278.3% |
> | Qwen-4B | 663.6% | 399.6% | 329.7% |
> | Qwen-8B | 614.2% | 229.3% | 181.7% |
> | Flan-UL2 | 421.5% | 257.0% | 962.1% |
> | DeepSeek-R1-70B | 566.5% | 448.1% | 388.7% |
>
> ②U8k4-W2: Lack of concrete, implementable intervention strategies.
>
> We appreciate this important suggestion. We therefore added a parameter-efficient fine-tuning strategy on critical heads. Using our coarse-to-fine framework, we fine-tune only the parameters associated with identified critical heads on a safety CoT dataset, while freezing the rest of the model. This updates only about 1% of parameters, but improves CoT safety by +67.1% on News and +55.0% on HarmBench on average. Meanwhile, the average drops on MATH500 and GPQA are only -1.5 and -1.7, showing that targeted head-level intervention can improve safety while largely preserving reasoning ability. We agree that spectral regularization is also promising, but prioritized this more direct intervention in the revision.
>
> | **Model** | **FT Ratio** | **News (Before / After)** | **HarmBench (Before / After)** |
> |---|---:|---:|---:|
> | Llama-8B | 1.56% | 21.1% / 94.7% | 20.8% / 87.4% |
> | Qwen-4B | 1.95% | 20.3% / 92.1% | 20.7% / 85.8% |
> | Qwen-8B | 1.43% | 19.6% / 90.2% | 31.2% / 84.6% |
> | Flan-UL2 | 1.03% | 45.2% / 88.4% | 41.6% / 82.9% |
> | DeepSeek-R1-70B | 0.64% | 10.5% / 86.7% | 34.9% / 83.7% |
> | **Avg. improvement** | - | **+67.1%** | **+55.0%** |
>
> | **Model** | **MATH500 (Before / After)** | **GPQA (Before / After)** |
> |---|---:|---:|
> | Llama-8B | 51.4% / 50.0% | 32.3% / 31.3% |
> | Qwen-4B | 89.2% / 87.8% | 65.7% / 63.9% |
> | Qwen-8B | 92.2% / 90.6% | 64.6% / 62.7% |
> | Flan-UL2 | 95.2% / 93.7% | 67.2% / 65.8% |
> | DeepSeek-R1-70B | 97.4% / 95.6% | 71.7% / 69.4% |
> | **Avg. change** | **-1.5%** | **-1.7%** |
>
> ③U8k4-W3: Clarification on related work and comparative analysis of baseline methods [KQ3/4]
>
> Thank you for this valuable suggestion. We added comparisons with two representative baselines: attention-based [1] and faithfulness-based [2] metrics. We compare both task correlation and runtime efficiency. Under our setup with 8×RTX 3090 GPUs, our Jacobian-based metrics add about 0.71s overhead on average, but outperform the baselines by about 59.0% in correlation with CoT safety. This indicates that our method is slightly more expensive, but more effective at capturing signals of unsafe CoT reasoning.
>
> | **Metric** | **News Datasets Corr. ↑** | **News Datasets Time ↓** | **HarmBench Corr. ↑** | **HarmBench Time ↓** |
> |---|---:|---:|---:|---:|
> | B1 | 0.88 | 1.45s | 0.82 | 1.52s |
> | B2 | 0.81 | 1.49s | 0.71 | 1.56s |
> | B3 | 0.73 | 1.71s | 0.89 | 1.78s |
> | Attention | 0.58 | 1.03s | 0.73 | 1.33s |
> | faithfulness | 0.38 | 0.54s | 0.64 | 0.60s |
>
> ④U8k4-W4: Mechanistic explanation for layer shift under indirect prompting.
>
> We appreciate this insightful question. Indirect prompts require the model to first infer hidden malicious intent from a superficially benign request before entering safety-critical reasoning. This extra inference step delays the divergence between safe and unsafe trajectories to deeper layers. This interpretation is also consistent with prior work showing that harmfulness and refusal are encoded as separate internal signals that emerge progressively across layers [3].
>
> [1]Zhou, et al. On the role of attention heads in large language model safety. In International Conference on Learning Representations (ICLR 2025).
>
> [2]Lanham, Tamera, et al. "Measuring faithfulness in chain-of-thought reasoning." arXiv preprint arXiv:2307.13702 (2023).
>
> [3]Zhao J, et al. LLMs Encode Harmfulness and Refusal Separately[C]//The Thirty-ninth Annual Conference on Neural Information Processing Systems(NeurIPS2025).

---

> > ### Author Rebuttal · Reviewer_U8k4 · 2026-04-01
> >
> > Thank you for the detailed rebuttal. The authors have comprehensively addressed all my original concerns with detailed supplementary experiments.

---

> > > ### Author Response · Authors · 2026-04-08
> > >
> > > We sincerely thank Reviewer U8k4 for the important questions on generalizability and scalability, which helped us further strengthen the empirical scope and practical relevance of the paper, and we will incorporate the relevant improvements into the revised version.

---

### Official Review · Reviewer_GtAy · 2026-03-06

**Soundness:** 3
**Presentation:** 4
**Significance:** 3
**Originality:** 3
**Overall Recommendation:** 4
**Confidence:** 2

**Summary:**

The central concept of this paper is the hidden safety risk within the Chain-of-Thought (CoT) reasoning processes of LLMs  in the context of fake news generation (FNG). Althought the LLM rejects the request for generating fake news, its CoT reasoning processes may provide some help.

Through extensive empirical analysis on models like Llama-8B and Qwen-4B/8B, the paper reveals that even when models output a safe refusal, approximately 80% of their internal CoT traces still contain actionable unsafe contents.

To analyze this, the paper proposes a unified, coarse-to-fine analytical framework. It first localizes "safety-critical layers" where safe and unsafe reasoning trajectories diverge. Then, it introduces a novel Jacobian-based spectral analysis method to identify specific "safety-critical attention heads." This method utilizes three interpretable metrics derived from the softmax Jacobian: Stability, Geometry , Energy. The authors validate their findings through causal interventions, showing that perturbing these critical heads significantly degrades model safety.

**Compliance With Llm Reviewing Policy:**

Affirmed.

**Key Questions For Authors:**

1. Does "safe output but unsafe CoT reasoning" exists in other malicious tasks beyond fake news generation?

2. Calculating the Jacobian matrix for attention heads can be computationally expensive. What is the computationl cost of this framework? Is this framework applicable to models larger than 8B?

**Limitations:**

yes

**Strengths And Weaknesses:**

Strengths

1. Novel and Critical Insight：This paper proposed a core idea that "output-level refusal does not imply safe CoT reasoning" and explore its influence in FNG, providing a new insight for studying LLM safety.

2. Rigorous Methodological Framework: The proposed coarse-to-fine analysis pipeline is well-structured. The transition from layer-level localization to head-level spectral analysis is logical and effective.

Weaknesses

This paper focuses solely on the  "safe output but unsafe CoT reasoning" phenomenon in fake news generation, lacking exploration of more other malicious tasks to demonstrate its universality.

---

> ### Author Rebuttal · Authors · 2026-03-31
>
> ①GtAy-W1 & KQ1: Exploration of malicious tasks
> We sincerely thank the reviewer for raising this important question. The answer is affirmative: the phenomenon of "safe output but unsafe CoT reasoning" is not exclusive to the fake news generation task. To address this concern, we conducted supplementary experiments on the jailbreak task in HarmBench, which covers diverse malicious tasks. Additionally, we expanded the model scope to include more large language models, such as Flan-UL2 and DeepSeek-R1-70B. Specifically, we adopted the definitions from the main text: the NN–NM gap denotes the average score difference between normal thinking trajectories (NN) and unsafe thinking trajectories (NM) within the same layer. Furthermore, Concentration is defined as the proportion of the critical layer gap to the total gap, while Relative Gap is defined as the ratio of the average gap of critical heads to that of non-critical heads within the critical layers. As shown in the table below, the discriminative signal on HarmBench is also primarily concentrated in a few critical layers, with Concentration ranging from 72.2% to 93.0% across different models. Meanwhile, within these critical layers, the gap on critical heads is also consistently significantly higher than that on non-critical heads. These results are consistent with the core findings in the main paper, namely that the divergence between safe and unsafe behaviors is not uniformly distributed across the entire network but is primarily concentrated in critical heads within a few critical layers. This demonstrates that the phenomenon is not specific to FNG but also exists in other malicious tasks.
> | Model | FT Ratio | News (Before / After) | HarmBench (Before / After) |
> |---|---|---|---|
> | Llama-8B | 1.56% | 21.1% / 94.7 | 20.8% / 87.4 |
> | Qwen-4B | 1.95% | 20.3% / 92.1 | 20.7% / 85.8 |
> | Qwen-8B | 1.43% | 19.6% / 90.2 | 31.2% / 84.6 |
> | Flan-UL2 | 1.03% | 45.2% / 88.4 | 41.6% / 82.9 |
> | DeepSeek-R1-70B | 0.64% | 10.5% / 86.7 | 34.9% / 83.7 |
> | **Avg. improvement** | - | **+67.1%** | **+55.0%** |
>
> | Model | MATH500 (Before / After) | GPQA (Before / After) |
> |---|---|---|
> | Llama-8B | 51.4 / 50.0 | 32.3 / 31.3 |
> | Qwen-4B | 89.2 / 87.8 | 65.7 / 63.9 |
> | Qwen-8B | 92.2 / 90.6 | 64.6 / 62.7 |
> | Flan-UL2 | 95.2 / 93.7 | 67.2 / 65.8 |
> | DeepSeek-R1-70B | 97.4 / 95.6 | 71.7 / 69.4 |
> | **Avg. change** | **-1.5** | **-1.7** |
>
> ②GtAy-W2 & KQ2：Calculation Cost
>
> In our analysis, B1, B2, and B3 are all built upon the same downsampled attention matrix. Let the downsampled matrix have size $n \times n$, where $n$ denotes the side length of the attention matrix; let $r$ denote the number of query rows involved in the operator analysis; and let $T$ denote the number of power iteration steps. For B1, the core operation is to perform $T$ rounds of power iteration on each selected row to approximate the dominant eigenvalue of the Jacobian, yielding a total complexity of $O(rTn)$. For B2, an additional linear post-processing step is applied to the resulting direction vector, so its complexity can be written as $O(rTn)+O(n)$. In contrast, B3 requires explicitly constructing the Jacobian matrix and further performing eigendecomposition to compute the spectral distribution or effective rank; the dominant cost of this step is the eigendecomposition, resulting in a complexity of $O(rn^3)$.
>
> As shown in the table below, we further compare the average runtime and task correlation of different metrics across multiple models. Under our experimental setup with eight NVIDIA RTX 3090 GPUs, the results show that although our metrics introduce an additional average overhead of approximately 0.71 seconds, they exhibit substantially stronger correlation(Corr.) with CoT-related tasks, outperforming the other metrics by an average margin of about 59.0%.
>
> | Metric | News Datasets Corr. ↑ | News Datasets Time ↓ | HarmBench Corr. ↑ | HarmBench Time ↓ |
> |---|---:|---:|---:|---:|
> | B1 | 0.88 | 1.45s | 0.82 | 1.52s |
> | B2 | 0.81 | 1.49s | 0.71 | 1.56s |
> | B3 | 0.73 | 1.71s | 0.89 | 1.78s |
> |Attention[1]| 0.58 | 1.03s | 0.73 | 1.33s |
> | Faithfulness[2] | 0.38 | 0.54s | 0.34 | 0.60s |
>
> [1]Zhou. et al. On the role of attention heads in large language model safety. In International Conference on Learning Representations (ICLR 2025).
>
> [2]Lanham, et al. "Measuring faithfulness in chain-of-thought reasoning." arXiv preprint arXiv:2307.13702 (2023).
>
> ③GtAy-W2：Applicability to models larger than 8B LLMs
>
> We sincerely thank the reviewer for this valuable suggestion and have actively supplemented validation on larger-scale models. Specifically, we have extended our experiments to include 20B and 70B parameter models. The results demonstrate that our core findings remain consistent on models larger than 8B, and the computational overhead remains within an acceptable range. These results indicate that our framework exhibits good scalability and can be effectively applied to larger-scale language models.

---

### Official Review · Reviewer_ZYN2 · 2026-03-11

**Soundness:** 3
**Presentation:** 2
**Significance:** 2
**Originality:** 4
**Overall Recommendation:** 5
**Confidence:** 3

**Summary:**

This work studies jailbreak behavior in LLMs when prompted to generate fake news, either directly or indirectly. The authors find that reasoning models operating in a “thinking” mode are more vulnerable to such attacks. Even when the final output is a refusal, the chain-of-thought (CoT) traces may still contain unsafe reasoning and reveal harmful strategies for responding to the malicious prompt. To analyze this phenomenon, the paper proposes a coarse-to-fine framework that first identifies safety-sensitive contiguous mid-depth layers using representation separation scores, and then applies spectral analysis of the softmax Jacobian to locate critical attention operations within those layers. Based on the Jacobian’s spectral properties, the authors define three metrics characterizing stability, geometric consistency, and energy concentration. Their experiments suggest that (1) safety-sensitive layers are concentrated in a small region of the network, and (2) safe reasoning is associated with more stable and focused routing, lower sensitivity, more consistent geometric alignment, and more concentrated energy than unsafe reasoning. The paper also studies variation across model architectures, news styles, and direct versus indirect prompting.

**Compliance With Llm Reviewing Policy:**

Affirmed.

**Final Justification:**

Thanks to the authors for resolving all my concerns. As they resolved all my questions, I have increased my overall score to a 5.

**Key Questions For Authors:**

1. In the last paragraph of Section 4.4, the perturbation analysis evaluates safety via a learned discriminator on final-layer representations. Why not use the generated CoT text itself to assess the perturbation effects on the safety of the actual generated outputs?

2. How would you mitigate the issues identified in this work? As stated before, the limits of this work can be addressed if either (A) fake-news generation were studied more deeply, through either a mitigation method or a finer-grained categorization of the impacts and formats of different types of fake-news generation, or (B) the method can be generalized beyond fake news generation to broader tasks such as illegal guidance/action suggestion, or aggressive, unethical, or racial content generation, i.e., broader jailbreaking behaviors. My rating could increase to 4 if concern (A) is addressed, or to 5 if concern (B) is addressed. Exploring both broader applications and a mitigation method would be an interesting future direction.

**Limitations:**

Yes

**Strengths And Weaknesses:**

*Strengths*

Soundness:
* The paper presents a technically interesting and reasonably well-motivated analysis framework. The combination of layer-level localization and attention-level analysis is coherent, and the paper supports its claims with both theoretical discussion and empirical results.
* The paper also examines the effect of different model architectures, prompting styles, and news-writing styles, improving the overall empirical depth of the study.

Originality:
* The work offers a novel perspective on CoT safety by connecting softmax Jacobian analysis with three physics-inspired metrics to study safety-sensitive routing behavior. This coarse-to-fine analysis pipeline is a creative way to localize safety-relevant components in reasoning models.

Empirical support:
* The experiments provide evidence for the paper’s two main hypotheses: (1) safety-sensitive behavior is concentrated in a relatively small subset of layers, and (2) safe reasoning is associated with more stable, focused, and geometrically consistent routing than unsafe reasoning.

*Weaknesses*

Significance:
* The empirical study is limited to fake-news generation, which is a relatively narrow application setting. While this is a meaningful safety use case, the impact of the findings may remain somewhat specialized unless the authors either demonstrate transfer to other harmful-generation tasks or derive more concrete practical implications for the fake-news setting (e.g., through a deeper analysis of different fake-news categories, a filtering or intervention strategy, or evidence that the findings can inform improved safety alignment for fake-news prompts).

Practical impact:
* Given the task-specific focus on fake-news generation, I expected the analysis to lead to at least one concrete mitigation, intervention, or defense direction for this setting. As written, the contribution is primarily diagnostic, which limits its practical impact.

Presentation:
* The repeated use of boxed “Key insight” summaries in the main text feels unnecessary; well-written sections should be self-explanatory. At the same time, the main framework overview (Fig. 8) would be more useful in the main paper than in the appendix.
* The caption of Fig. 1 is awkwardly written. In particular, the final phrase (“surface-level refusal”) reads like a dangling fragment and should be revised for clarity.
* In Section 4.1, line 173, the “K” in “While K localizes critical layers…” appears to have inconsistent font styling compared with earlier occurrences, which may be a minor typo issue.

---

> ### Author Rebuttal · Authors · 2026-03-31
>
> ①ZYN2-W1, **Concern (B)**: Generalize to other safety domains
>
> We sincerely thank the reviewer for valuable suggestions regarding the scope of the empirical analysis and its practical impact. To address this concern, we have conducted supplementary experiments on the HarmBench dataset (Jailbreak Task) and expanded the model scope to include additional large language models, including Flan-UL2(20B) and DeepSeek-R1-70B. As shown in the table below, the discriminative signal, measured by the score difference between NN and NM, is primarily concentrated in a few critical layers (Concentration =Average Critical Layer Gap /Average Total Gap). Furthermore, within these layers, the mean NN–NM gap on critical heads is also significantly higher than the mean of other heads (Relative-Gap = Mean Gap(Critical-Layers) / Mean Gap(Non-Critical Layers)). These results confirm that safe/unsafe behavior divergence concentrates in critical heads within critical layers.
>
> | Model/HarmBench | Layer Range | Conc. |
> |---|---|---|
> | Llama-8B | [56.2%, 62.5%] | 72.2% |
> | Qwen-4B | [86.1%, 91.7%] | 82.8% |
> | Qwen-8B | [75.0%, 80.6%] | 73.9% |
> | Flan-UL2 | [80.6%, 86.9%] | 93.0% |
> | DeepSeek-R1-70B | [82.5%, 85.0%] | 78.1% |
>
> | Model | B1 Relative Gap | B2 Relative Gap | B3 Relative Gap |
> |---|---:|---:|---:|
> | Llama-8B | 321.4% | 322.6% | 278.3% |
> | Qwen-4B | 663.6% | 399.6% | 329.7% |
> | Qwen-8B | 614.2% | 229.3% | 181.7% |
> | Flan-UL2 | 421.5% | 257.0% | 962.1% |
> | DeepSeek-R1-70B | 566.5% | 448.1% | 388.7% |
>
> ②ZYN2-W2, **Concern (A)**: Lack of mitigation strategies & practical impact.
>
> We sincerely appreciate the reviewer's critical feedback regarding practical impact. To address the concern that "the work remains primarily diagnostic without further mitigation," we have developed a critical-head-based fine-tuning strategy. Specifically, leveraging our "coarse-to-fine" analysis framework, we first localize the intermediate layers most critical to safety, and then identify the critical attention heads within them using three Jacobian-based metrics: stability, geometry, and energy (methods and formulas are detailed in the submitted paper). Based on this, we collected safe responses across different task types to construct a prompt-CoT safety dataset, and fine-tuned only a small subset of parameters in the critical heads while freezing the remaining parameters. Experiments demonstrate that this method requires updating only approximately 1% of parameters yet significantly improves CoT safety, with an average improvement of approximately 60%. Meanwhile, results on MATH500 and GPQA indicate that this alignment has a minimal negative impact (-1.6%) on the model's reasoning capabilities.
>
> | Model | FT Ratio | News (Before / After) | HarmBench (Before / After) |
> |---|---|---|---|
> | Llama-8B | 1.56% | 21.1% / 94.7 | 20.8% / 87.4 |
> | Qwen-4B | 1.95% | 20.3% / 92.1 | 20.7% / 85.8 |
> | Qwen-8B | 1.43% | 19.6% / 90.2 | 31.2% / 84.6 |
> | Flan-UL2 | 1.03% | 45.2% / 88.4 | 41.6% / 82.9 |
> | DeepSeek-R1-70B | 0.64% | 10.5% / 86.7 | 34.9% / 83.7 |
> | **Avg. improvement** | - | **+67.1%** | **+55.0%** |
>
> | Model | MATH500 (Before / After) | GPQA (Before / After) |
> |---|---|---|
> | Llama-8B | 51.4 / 50.0 | 32.3 / 31.3 |
> | Qwen-4B | 89.2 / 87.8 | 65.7 / 63.9 |
> | Qwen-8B | 92.2 / 90.6 | 64.6 / 62.7 |
> | Flan-UL2 | 95.2 / 93.7 | 67.2 / 65.8 |
> | DeepSeek-R1-70B | 97.4 / 95.6 | 71.7 / 69.4 |
> | **Avg. change** | **-1.5** | **-1.7** |
>
> ③ZYN2-W3：Writing Presentation
>
> We are grateful for your detailed review of writing. Regarding the "Key insight" boxes, we agree that they are somewhat redundant and will remove them in the revised manuscript, integrating the core content into the submitted paper. We also accept your suggestion regarding Fig.8 and will move it from the appendix to the beginning of Section 4 to help readers better understand the overall framework. In addition, the oversight in the caption of Fig.1 concerning the phrase "surface-level refusal" has been corrected to make the caption clearer and more complete. Finally, the inconsistent font styling of the mathematical symbol "K" in Section 4.1 will also be unified in the revised manuscript.
>
> ④ZYN2-W4: Perturbation analysis evaluation
>
> We sincerely thank the reviewer for raising this question. We did not directly evaluate safety based on the perturbed CoT text because internal perturbations often significantly disrupt generation stability, leading to semantic fragmentation, repetition, or gibberish in the output. In such cases, safety judgments at the text level often confound changes in safety with generation degradation, making them unreliable indicators. In contrast, evaluating using a discriminator on final-layer representations within the state space allows us to more directly characterize the impact of perturbations on safety-related internal representations, thereby avoiding the confusion introduced by textual distortion.

---

> > ### Author Rebuttal · Reviewer_ZYN2 · 2026-04-02
> >
> > Thanks to the authors for resolving all my concerns.

---

> > > ### Author Response · Authors · 2026-04-08
> > >
> > > We sincerely thank Reviewer ZYN2 for the valuable comments on practical impact, broader significance, and presentation, which helped improve the clarity and completeness of the paper, and we will incorporate them into the revised version.

---

### Official Review · Reviewer_eDia · 2026-03-13

**Soundness:** 4
**Presentation:** 4
**Significance:** 4
**Originality:** 4
**Overall Recommendation:** 5
**Confidence:** 4

**Summary:**

This paper investigates a critical, overlooked vulnerability in reasoning-oriented Large Language Models (LLMs): the assumption that a model's refusal to generate harmful content implies its internal reasoning process is also safe. Focusing on fake news generation, the authors introduce a unified mechanistic interpretability framework that deconstructs Chain-of-Thought (CoT) generation across model layers. By employing Jacobian-based spectral metrics, the study introduces three interpretable measures: stability, geometry, and energy, to quantify how individual attention heads process deceptive narratives. The key empirical finding reveals that even when an LLM ultimately rejects a harmful prompt, its internal CoT may still generate and propagate unsafe narratives, with critical routing decisions heavily concentrated in a few contiguous mid-depth layers.

**Compliance With Llm Reviewing Policy:**

Affirmed.

**Key Questions For Authors:**

Q1: Given that you have successfully localized the critical routing decisions to specific mid-depth layers, have you attempted any targeted interventions (e.g., activation steering, head ablation) to actively suppress the deceptive reasoning patterns? If so, how did it impact the model's overall reasoning capabilities?

Q2: Do the structural findings (i.e., the specific mid-depth layer concentration) hold true for other types of malicious prompts (e.g., typical jailbreaks or harmful instructions), or is this architecture specifically tuned to the linguistic nuances of fake news?

Q3: Could you provide a formal complexity analysis of computing the Jacobian-based metrics? Specifically, how does the computational overhead scale with the number of tokens in the CoT, and is there any potential to optimize this for real-time, online safety monitoring?

Q4: Does the length of the reasoning process exacerbate the generation risk? Is there a measurable correlation between the number of thinking tokens generated and the spectral energy of the unsafe narratives embedded in the CoT?

**Limitations:**

yes

**Strengths And Weaknesses:**

Strengths
1. As the field shifts toward test-time compute and reasoning models (e.g., OpenAI o-series, DeepSeek-R1), evaluating safety solely via final output is becoming obsolete. Challenging the "refusal implies safe reasoning" assumption is a highly timely and significant contribution to AI alignment, exposing a hidden surface for latent risks.

2. Moving beyond shallow behavioral evaluations, the paper roots its analysis in mechanistic interpretability. The introduction of Jacobian-based spectral metrics to derive physical/mathematical analogs (stability, geometry, energy) provides a mathematically rigorous, white-box lens to evaluate attention head dynamics during CoT.

3. The empirical localization of deceptive reasoning patterns to "contiguous mid-depth layers" is a striking finding. This granular identification provides a highly specific target for future safety interventions, such as representation engineering or targeted ablation, avoiding the need for full-model fine-tuning.

Weaknesses

1. While the paper excels as a diagnostic tool, it appears to stop short of proposing a concrete mitigation strategy. For a top-tier ML venue, reviewers typically look for the closed loop: identifying the vulnerability (which this paper does well) and demonstrating a method to patch it (e.g., activation steering on the identified mid-depth heads to suppress the unsafe CoT without degrading general reasoning).

2. The empirical analysis is strictly scoped to "Fake News Generation." It remains unclear whether the observed phenomena, specifically the concentration of risk in mid-depth layers, generalize to other critical safety domains, such as jailbreaks, malicious code generation, or bio-terrorism planning.

3. Computing Jacobian-based spectral metrics for individual attention heads across all layers over long CoT sequences involves heavy matrix operations. The paper lacks a clear discussion on the computational complexity of this framework. It is unclear if this can scale to models with 100B+ parameters or if it is strictly an offline, small-scale diagnostic tool.

---

> ### Author Rebuttal · Authors · 2026-03-31
>
> ① eDia-W1 & Q1: Lack of a concrete mitigation strategy and impact of targeted interventions.
>
> Thank you for this important suggestion on closing the diagnosis-to-mitigation loop. We develop a targeted mitigation strategy based on critical heads identified by our framework: localizing safety-critical mid-depth layers, then using our three Jacobian-based metrics (stability, geometry, energy) to identify the most safety-relevant attention heads. We construct a prompt–CoT safety fine-tuning dataset from safe responses across multiple task types, and fine-tune only the ~1% parameters associated with these key heads while freezing others. This improves CoT safety by +67.1% on News and +55.0% on HarmBench on average, with minimal reasoning drops (−1.5 on MATH500, −1.7 on GPQA), confirming these key heads as both diagnostically meaningful and effective intervention targets for risk localization → targeted mitigation.
>
> | Model | FT Ratio | News (Before / After) | HarmBench (Before / After) |
> |---|---:|---:|---:|
> | Llama-8B | 1.56% | 21.1% / 94.7% | 20.8% / 87.4% |
> | Qwen-4B | 1.95% | 20.3% / 92.1% | 20.7% / 85.8% |
> | Qwen-8B | 1.43% | 19.6% / 90.2% | 31.2% / 84.6% |
> | Flan-UL2 | 1.03% | 45.2% / 88.4% | 41.6% / 82.9% |
> | DeepSeek-R1-70B | 0.64% | 10.5% / 86.7% | 34.9% / 83.7% |
> | **Avg. improvement** | - | **+67.1%** | **+55.0%** |
>
> | Model | MATH500 (Before / After) | GPQA (Before / After) |
> |---|---:|---:|
> | Llama-8B | 51.4% / 50.0% | 32.3% / 31.3% |
> | Qwen-4B | 89.2% / 87.8% | 65.7% / 63.9% |
> | Qwen-8B | 92.2% / 90.6% | 64.6% / 62.7% |
> | Flan-UL2 | 95.2% / 93.7% | 67.2% / 65.8% |
> | DeepSeek-R1-70B | 97.4% / 95.6% | 71.7% / 69.4% |
> | **Avg. change** | **-1.5%** | **-1.7%** |
>
> ② eDia-W2 & Q2: Generalization beyond fake news generation.
>
> We appreciate this important question on generalization To address it, we extend our experiments from fake news generation to the broader harmful-instruction setting using HarmBench, and further include additional models with different scales and architectures, including Flan-UL2 and DeepSeek-R1-70B. The same structural pattern generalizes beyond fake news: safety-discriminative signals (NN–NM score gap) remain highly concentrated in a small set of mid-depth layers, where identified key heads exhibit substantially larger average gaps than non-key heads. This indicates our findings reflect a general mechanism: under diverse malicious prompts, safe and unsafe reasoning trajectories diverge primarily within a limited set of critical mid-depth layers and heads.
>
> | Model | HarmBench Relative Layer Range | HarmBench Conc. |
> |---|---:|---:|
> | Llama-8B | [56.2%, 62.5%] | 72.2% |
> | Qwen-4B | [86.1%, 91.7%] | 82.8% |
> | Qwen-8B | [75.0%, 80.6%] | 73.9% |
> | Flan-UL2 | [80.6%, 86.9%] | 93.0% |
> | DeepSeek-R1-70B | [82.5%, 85.0%] | 78.1% |
>
> | Model | B1 Relative Gap | B2 Relative Gap | B3 Relative Gap |
> |---|---:|---:|---:|
> | Llama-8B | 321.4% | 322.6% | 278.3% |
> | Qwen-4B | 663.6% | 399.6% | 329.7% |
> | Qwen-8B | 614.2% | 229.3% | 181.7% |
> | Flan-UL2 | 421.5% | 257.0% | 962.1% |
> | DeepSeek-R1-70B | 566.5% | 448.1% | 388.7% |
>
> ③eDia-W3 & Q3: Computational complexity and scalability of the Jacobian-based framework.
>
> We thank the reviewer for this important question. For a single downsampled attention matrix of size $n \times n$, with $r$ query rows and $T$ power-iteration steps, the time complexities of B1, B2, and B3 are $O(rTn)$, $O(rTn) + O(n)$, and $O(rn^3)$, respectively, with B3 typically dominating the total cost. Empirically, compared with attention-based [1] and faithfulness-based [2] baselines, our method adds only about $0.71\text{s}$ average overhead on $8 \times \text{RTX 3090}$, while achieving about $59.0\%$ higher average correlation with CoT safety tasks. Since the relevant signals are concentrated in a small set of key layers and heads, the framework can also be reduced to compute only on the critical positions. Though not yet fully evaluated on models beyond $100\text{B}$ parameters, our method is not inherently limited to small-scale offline diagnosis and shows clear potential for scalable, efficient online monitoring.
>
> [1]Zhou. et al. On the role of attention heads in large language model safety. In International Conference on Learning Representations (ICLR 2025).
>
> [2]Lanham, et al. "Measuring faithfulness in chain-of-thought reasoning." arXiv preprint arXiv:2307.13702 (2023).
>
> ④eDia-W4 & Q4: Correlation between reasoning length and spectral energy.
> We appreciate this insightful question. Per our three metrics' definitions, longer CoTs do amplify generation risk: using 0–1500 tokens as reference, CoTs of 1500–2000 tokens consistently show higher B1, higher B2, and lower B3 across all five models.
>
> | Model | B1 (1500–2000) | B2 (1500–2000) | B3 (1500–2000) |
> |---|---:|---:|---:|
> | Llama-8B | +30.8% | +374.7% | -46.8% |
> | Qwen-4B | +32.1% | +117.2% | -5.5% |
> | Qwen-8B | +19.2% | +69.8% | -17.1% |
> | Flan-UL2 | +42.5% | +18.4% | -12.5% |
> | DeepSeek-R1-70B | +7.9% | +13.7% | -32.7% |

---

> > ### Author Rebuttal · Reviewer_eDia · 2026-04-02
> >
> > Thank you for your explanation.

---

> > > ### Author Response · Authors · 2026-04-08
> > >
> > > We sincerely thank Reviewer eDia for the thoughtful feedback on mitigation, generalization, and computational practicality, which helped us strengthen the paper, and we will incorporate these improvements into the revised version.

---

### Decision · Program_Chairs · 2026-04-30

**Decision:**

Accept (regular)

**Comment:**

This paper studies an important safety issue in LLMs: even when models refuse harmful requests, their internal CoT reasoning can still contain unsafe content. Reviewers found this insight both interesting and important, and viewed the method as technically strong and well executed.

The initial concerns were fairly minor (scope, lack of mitigation, and computational cost). The rebuttal clearly improved the paper: the authors added experiments showing the results generalise beyond fake news, introduced a concrete mitigation method, and clarified efficiency. These additions made the contribution much more complete.  All reviewers stated that their concerns were fully resolved after the rebuttal and gave strong final scores. I also found the paper enjoyable to read, with a clear motivation and a compelling combination of empirical insights and mechanistic analysis, a clear contribution for the ICML community. Overall, I recommend acceptance.